# LSPT: Long-term Spatial Prompt Tuning for Visual Representation Learning

## Abstract

Visual Prompt Tuning (VPT) techniques have gained prominence for their capacity to adapt pre-trained Vision Transformers (ViTs) to downstream visual tasks using specialized learnable tokens termed as prompts. Contemporary VPT methodologies, especially when employed with self-supervised vision transformers, often default to the introduction of new learnable prompts or gated prompt tokens predominantly sourced from the model's previous block. A pivotal oversight in such approaches is their failure to harness the potential of long-range previous blocks as sources of prompts within each self-supervised ViT. To bridge this crucial gap, we introduce Long-term Spatial Prompt Tuning (LSPT) – a revolutionary approach to visual representation learning. Drawing inspiration from the intricacies of the human brain, LSPT ingeniously incorporates long-term gated prompts. This feature serves as temporal coding, curbing the risk of forgetting parameters acquired from earlier blocks. Further enhancing its prowess, LSPT brings into play patch tokens, serving as spatial coding. This is strategically designed to perpetually amass class-conscious features, thereby fortifying the model's prowess in distinguishing and identifying visual categories. To validate the efficacy of our proposed method, we engaged in rigorous experimentation across 5 FGVC and 19 VTAB-1K benchmarks. Our empirical findings underscore the superiority of LSPT, showcasing its ability to set new benchmarks in visual prompt tuning performance.

## 1 Introduction

The rise of the Transformer architecture (Vaswani et al., 2017) has cemented its position as the foundational module for vision-related tasks. Within this paradigm, Vision Transformers (ViTs) (Dosovitskiy et al., 2021; Touvron et al., 2020; Liu et al., 2021; Yuan et al., 2021) have manifested remarkable dominance over traditional Convolutional Neural Networks (CNNs) across various tasks, such as image classification, object detection, and semantic segmentation. Concurrently, the success of self-supervised learning frameworks(Chen et al., 2020; Chen & He, 2021; He et al., 2020; Grill et al., 2020), especially in harnessing vast reservoirs of unlabeled data, has been undeniable. Merging these two powerhouses seems instinctual, and early forays into this combination (Chen et al., 2021; Xie et al., 2021; Caron et al., 2021) indeed show promise, despite challenges in seamless integration.

Amidst this backdrop, Visual Prompt Tuning (VPT) has emerged as an influential player, adept at tailoring pre-trained ViTs for specific downstream tasks using adaptable tokens or "prompts". As a testament to VPT's prowess, VPT techniques prepend learnable prompts to input sequences, effectively guiding the fixed pre-trained encoder's information for task-specific objectives (Jia et al., 2022). The Gated Prompt Tuning (GaPT) strategy takes this a notch further by incorporating gating mechanisms for each ViT block to modulate its influence based on cues from preceding blocks (Yoo et al., 2023).

Nevertheless, some glaring limitations remain. Firstly, these methodologies largely overlook the latent potential of long-range blocks as prompt sources among blocks and cause the *temporal* forgetting problem , which also corresponds to "long-term" forgetting across transformer blocks. Even with the gating mechanism introduced in the gated prompt tuning strategy, the information from early blocks diminishes exponentially, making it challenging to capture in the later blocks. Additionally, the embedding of patch tokens, which encapsulates crucial spatial information and acts as

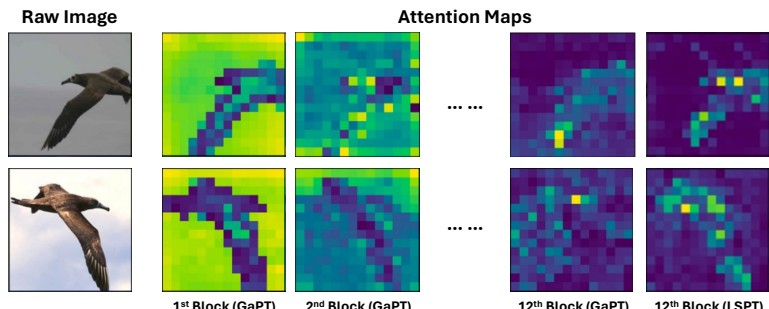

Figure 1: Comparison of the forgetting problem in GaPT and the shape information awareness in our LSPT. For the 12th block, the attention map of the state-of-the-art approach has been blur and almost lose the crucial spatial information. While for our LSPT, we can see a clear attention map for the object in the raw image, demonstrating its ability to incorporate spatial information and pass it through long-range blocks.

an intermediate global visual representation of the image, is regrettably lost across blocks. This results in what can be termed as a *spatial* forgetting phenomenon. Figure 1 illustrates the nature of the forgetting issue inherent in contemporary visual prompt tuning methods. Intriguingly, these two feature forgetting challenges align remarkably with the human visual system. In humans, both temporal and spatial correlations of neuronal discharges are essential for integrating distributed neuronal activities into cohesive representations (Huxter et al., 2003; Victor & Purpura, 1996; Engel et al., 1992; Reinagel & Reid, 2000). Therefore, we posit that integrating both temporal and spatial coding could significantly enhance the efficacy of visual prompts.

To address these pressing concerns, we unveil the Long-term Spatial Prompt Tuning (LSPT) framework that can explicitly alleviate the forgetting issues on both temporal and spatial aspects. Rooted deeply in neural mechanisms found in the human brain, LSPT offers a fresh perspective to visual representation learning. At its core, LSPT integrates long-term gated prompts, introducing a temporal coding layer that actively mitigates the forgetting of parameters learned from anterior blocks. By weaving in patch tokens as spatial coding elements, it additionally ensures a sustained aggregation of class-specific features, bolstering the model's discriminative capabilities. Subjecting LSPT to meticulous evaluations on 5 FGVC and 19 VTAB-1K benchmarks, we unearth empirical evidence attesting to its unparalleled prowess, setting novel standards in visual prompt tuning.

In a nutshell, our seminal contributions are:

- The inception of LSPT: a pioneering prompt tuning paradigm adept at seamlessly integrating long-term gated prompts for temporal coding, effectively addressing the 'forgetting' challenges of preceding approaches.
- The novel integration of learnable spatial gated prompts, meticulously crafted to ensure a continuous accumulation of class-distinctive features.
- Comprehensive experimental validations that unequivocally establish LSPT's supremacy over existing baselines in the realm of visual prompt tuning.

## 2 RELATED WORK

**Self-supervised Vision Transformers** (Chen et al., 2021; Xie et al., 2021; Caron et al., 2021) have addressed people's attention due to their strong performance on various downstream tasks. Specifically, MoCov3 (Chen et al., 2021) extended the MoCo (He et al., 2020) method to ViT (Dosovitskiy et al., 2021) for minimizing the distance between representations of two augmented views. MoCo v2 and BYOL were applied simultaneously in MOBY (Xie et al., 2021) to form a self-supervised framework based on the Swin (Liu et al., 2021) backbone. In DINO (Caron et al., 2021), knowledge distillation was combined with momentum encoder and multi-crop training for learning the local-to-global correspondence in the vision transformer. As proven to be effective in a previous study (Raghu et al., 2021), vision transformers can obtain global representations from shallow layers. Therefore, it is desirable to take into account low-level features from the shallow stage for

learning more fine-grained invariances. Masked image modeling (MIM) also has been explored in many self-supervised ViTs (Bao et al., 2021; Atito et al., 2021; He et al., 2021; Wei et al., 2022; Xie et al., 2022) to reconstruct the masked image patch given the unmasked counterpart as clues. For example, block-wise masking was introduced in BEiT (Bao et al., 2021) to learn transferrable visual representations by recovering discrete tokens of masked image patches. Given features extracted from the 25% unmasked patches, the seminal work, MAE (He et al., 2021) directly reconstructed missing pixels of 75% masked patches. In this work, our main focus is to adapt self-supervised pre-trained vision transformers to downstream visual tasks using specialized learnable prompts, which is more challenging than fine-tuning all parameters of the pre-trained backbone architecture.

**Visual Transfer Learning** aims to learn transferable representations from pre-trained vision backbones for downstream tasks. Early works Dosovitskiy et al. (2021) leveraged full fine-tuning to train both the pre-trained model and the task-specific head. Recently, diverse parameter-efficient tuning methods have been proposed to For example, Sidetune (Zhang et al., 2020) utilized a "side" network and linearly interpolated between pre-trained features and side-tuned features before being fed into a classification head. Bias tuning (Cai et al., 2020; Ben Zaken et al., 2022) proposed to fine-tune only the bias terms of the pre-trained backbone. Adapter-based approaches (Houlsby et al., 2019; Pfeiffer et al., 2020a;b) inserted multiple MLP modules with residual connection inside visual transformer layers. However, since these mainly deal with supervised pre-trained ViTs, few studies explored parameter-efficient tuning for self-supervised models. In this work, we develop a new prompt-based transfer learning method based on learnable input prompts for self-supervised ViTs.

**Visual Prompt Tuning** (VPT) (Jia et al., 2022) prepended learnable prompt tokens to the input sequences, which then act as task-specific instructions by steering the information from the fixed pre-trained encoder. VPT, when used with supervised ViT backbones, has shown outstanding performance on numerous downstream tasks. GaPT (Yoo et al., 2023) proposed to adapt a gate for each ViT block to adjust its intervention into the prompt tokens predominantly sourced from the model's previous block. However, a pivotal oversight in those approaches is their failure to harness the potential of long-range previous blocks as sources of prompts within each self-supervised ViT. In contrast, we develop a fully novel framework to mitigate the forgetting of previously learned prompts from history transformer blocks with explicit long-term prompts and class-aware spatial prompt coding. To the best of our knowledge, we are the first to leverage an explicit temporal and spatial prompts coding mechanism for visual prompt tuning. Our experiments in Section 4.2 also validate the superiority of our LSPT in all benchmarks for prompt tuning.

## 3 METHOD

Given a set of images, our target is to efficiently adapt pre-trained Vision Transformers (ViTs) to downstream visual tasks using specialized learnable prompts. We propose a novel brain-inspired prompt tuning framework, named LSPT, for capturing long-range blocks as prompt sources within self-supervised ViTs, which mainly consists of two modules, Class-aware Spatial Prompt Coding in Section 3.2 and Long-term Prompt Coding in Section 3.3.

### 3.1 PRELIMINARIES

In this section, we first describe the problem setup and notations and then revisit the visual prompt tuning for downstream image classification.

#### 3.1.1 PROBLEM SETUP AND NOTATIONS

Given a set of downstream images, our goal is to adapt pre-trained Vision Transformers to downstream visual tasks using learnable prompt tokens. We have a ViT consisting of a patch embedding layer, a stack of $L$ transformer blocks, and a classification head. For an input image $\mathbf{I}$ with shape of $H \times W \times 3$, we denote the input patch tokens for the $l$-th block as $\mathbf{x}^{l-1} = [\mathbf{X}_1^{l-1}, ..., \mathbf{X}_N^{l-1}] \in \mathbb{R}^{N \times D}$, where $N = HW/P^2, l = 1, ..., L$, $P$ is the patch size, and $D$ is the dimension of the transformer blocks. $\mathbf{X}_i^0 = \text{embed}(\mathbf{x}_i), i \in \{1, 2, ..., N\}$ is obtained by embedding the $i$-th patch $\mathbf{x}_i$ of the input image $\mathbf{I}$. An additional learnable classification token $\mathbf{X}_C^{l-1} \in \mathbb{R}^{1 \times D}$ is also concatenated to

patch tokens for each self-attention block, that is, $[\mathbf{X}_C^l, \mathbf{X}^l] = \text{AttnBlock}^l([\mathbf{X}_C^{l-1}, \mathbf{X}^{l-1}])$, where $\mathbf{X}_C^l \in \mathbb{R}^{1 \times D}, \mathbf{X}_C^l \in \mathbb{R}^{N \times D}$.

### 3.1.2 REVISIT VISUAL PROMPT TUNING

To solve the prompt tuning problem for visual classification, VPT (Jia et al., 2022) proposed to fine-tune continuous prompt tokens directly in the representation space and prepended these prompt tokens $\mathbf{P} = [\mathbf{p}_1, ..., \mathbf{p}_{N_p}] \in \mathbb{R}^{N_p \times D}$ to the input patch tokens, where $N_p$ is the number of learnable prompt tokens and $D$ is the dimension size of the prompt tokens shared with patch tokens. Specifically, they froze the pre-trained ViT weights and fine-tuned the newly inserted prompt tokens $\mathbf{P}$ and a classification head for specific downstream tasks. The first variant called VPT-shallow is to insert prompt tokens $\mathbf{P}$ as input only in the first block, which is formulated as

$$[\mathbf{x}_C^l, \mathbf{X}_P^l, \mathbf{X}^l] = \text{AttnBlock}^l([\mathbf{x}_C^{l-1}, \mathbf{P}, \mathbf{X}^{l-1}]), \quad l = 1 \tag{1}$$

$$[\mathbf{x}_C^l, \mathbf{X}_P^l, \mathbf{X}^l] = \text{AttnBlock}^l([\mathbf{x}_C^{l-1}, \mathbf{X}_P^{l-1}, \mathbf{X}^{l-1}]), \quad l = 2, ..., L \tag{2}$$

While the aggregated information from prompt tokens and patch tokens are passed along all blocks, the size of learnable prompts severely limits the transferring capability. In order to incorporate more tuning prompts in the embedding space, VPT-deep tried to inject new block-specific prompt tokens $\mathbf{P}^{l-1} = [\mathbf{p}_1^{l-1}, ..., \mathbf{p}_{N_p}^{l-1}] \in \mathbb{R}^{N_p \times D}$ to each block, which is formulated as

$$[\mathbf{x}_C^l, \mathbf{X}_P^l, \mathbf{X}^l] = \text{AttnBlock}^l([\mathbf{x}_C^{l-1}, \mathbf{P}^{l-1}, \mathbf{X}^{l-1}]), \quad l = 1, ..., L \tag{3}$$

Note that $\mathbf{X}_P^l$ will be discarded after each block, and will not be used as input to the next block. When training future blocks based on the output from current blocks, aggregated prompt tokens from the previous blocks might not be seen anymore.

However, such a visual prompt tuning setting will pose the main challenge for visual transformers to continually aggregate the newly injected prompt tokens $\mathbf{P}^{l-1}$ for the $l$-th block. The global visual representation extracted from the image is catastrophically forgotten by the stack of new prompt tokens, and thus they can not associate the latest prompts with the corresponding objects in the image for future blocks. Meanwhile, they ignored the explicit incorporation of patch token embeddings and learnable prompt tokens during training, causing worse attention maps from the last transformer layer. To tackle the challenge, we propose a novel prompt tuning framework, namely LSPT, for leveraging the latent potential of long-range blocks as prompt sources and class-aware spatial prompt coding to achieve efficient tuning within self-supervised ViTs, as illustrated in Figure 2.

### 3.2 CLASS-AWARE SPATIAL PROMPT CODING

To summarize from the two different designs of VPT-shallow and VPT-deep in Equation 2 and Equation 3, the key idea is to obtain an informative prompt input $\mathbf{X}_P^{l-1}$ for the $l$-th block. Meaningful sources for constructing the input are 3-fold: 1) the newly injected prompt token $\mathbf{P}^{l-1}$; 2) the output prompts which contain information from anterior blocks; 3) the output patch tokens which contain spatial information of the image tokens. The first source provides learnable parameters to enhance model's expressiveness during transferring and the last two offer valuable information which helps address the temporal and spatial forgetting problem.

We first introduce a novel and explicit class-aware spatial prompt coding module to incorporate global visual representation extracted from the image learned from previous self-attention blocks. Given output prompts $\widehat{\mathbf{X}}_P^{l-1} \in \mathbb{R}^{N_p \times D}$ and patch embeddings $\mathbf{X}^{l-1} \in \mathbb{R}^{N \times D}$ from $l-1$th transformer block, we add the average embedding of patch tokens to the $N_p$ output prompts for spatial prompt coding, which is formulated as

$$[\mathbf{x}_C^l, \widehat{\mathbf{X}}_P^l, \mathbf{X}^l] = \text{AttnBlock}^l([\mathbf{x}_C^{l-1}, \mathbf{X}_P^{l-1}, \mathbf{X}^{l-1}]), \quad l = 2, ..., L, \tag{4}$$

$$\widehat{\mathbf{X}}_{SP,k}^l = \widehat{\mathbf{X}}_{P,k}^l + \frac{\sum_{i=1}^N \mathbf{X}_i^l}{N}, \quad k = 1, 2, ..., N_p, \tag{5}$$

where $N_p, N$ denotes the number of learnable prompt tokens and embedded patches, respectively. The class-aware spatial prompt tokens $\widehat{\mathbf{X}}_{SP,k}^l \in \mathbb{R}^{N_p \times D}$ is further utilized to construct the input visual prompt tokens for the next block. It is noteworthy that this averaging operation helps the model

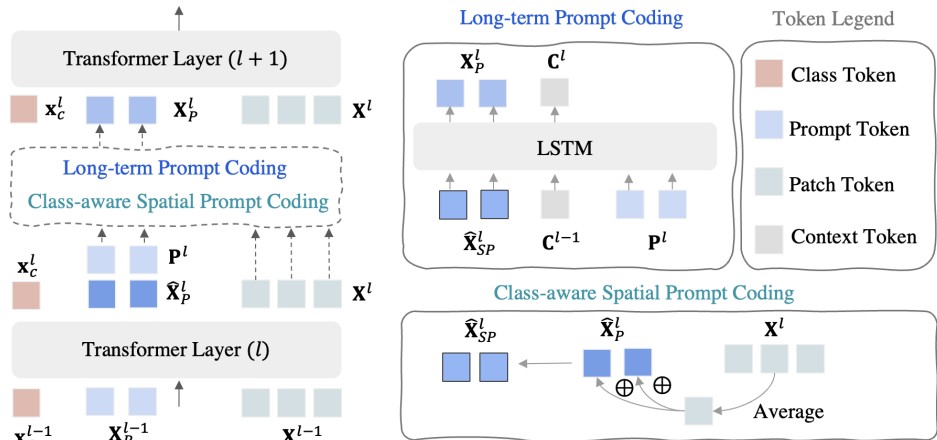

Figure 2: Illustration of the proposed Long-term Spatial Prompt Tuning (LSPT) framework. For transformer block $l$, the Class-aware Spatial Prompt Coding (CSPC) module adds the average embeddings of patch tokens $\mathbf{X}^l \in \mathbb{R}^{N \times D}$ from the block to the output prompts $\widehat{\mathbf{X}}_P^l \in \mathbb{R}^{N_p \times D}$ to generate class-aware spatial prompts $\widehat{\mathbf{X}}_{SP}^l$. With the inserted prompt tokens $\mathbf{P}^l \in \mathbb{R}^{N_p \times D}$ and $\widehat{\mathbf{X}}_{SP}^l \in \mathbb{R}^{N_p \times D}$, the Long-term Prompt Coding (LPC) module with parallel importance takes the inserted prompts $\mathbf{P}^l$ as input and $\widehat{\mathbf{X}}_{SP}^l$ as hidden states, and the output context embeddings $\mathbf{C}^{l-1} \in \mathbb{R}^{N_p \times D}$ at block $l-1$ are fed into the layer as cell states. Finally, the output updated prompts $\mathbf{X}_P^l$ is used as the new prompt tokens for block $l+1$ to achieve long-term prompt coding.

aggregate prompt tokens with explicit class-specific features and does not introduce any additional trainable parameters during transfer learning.

## 3.3 LONG-TERM PROMPT CODING

To explicitly learn from prompt sources across long-range blocks, we introduce a novel long-term prompt coding mechanism to mitigate the forgetting of previously learned prompts corresponding to objects in the image. Specifically, we leverage a learnable temporal coding layer consisting of long short-term memory (LSTM) (Hochreiter & Schmidhuber, 1997) with dimension size of $D$ to avoid forgetting the spatial prompt tokens $\widehat{\mathbf{X}}_{SP}^l$ introduced in the previous blocks while injecting new learnable prompt tokens $\mathbf{P}^l$.

For LSTM, given an input at current step $t$ and the hidden state $h_{t-1}$ and context state $c_{t-1}$ at time step $t-1$, the final output with hidden state $h_t$ and context state $c_t$ is defined as

$$h_t, c_t = \text{LSTM}(h_{t-1}, c_{t-1}, x_t) \tag{6}$$

With the inserted prompt tokens $\mathbf{P}^l \in \mathbb{R}^{N_p \times D}$ and $\widehat{\mathbf{X}}_{SP}^l \in \mathbb{R}^{N_p \times D}$, we take the inserted prompts $\mathbf{P}^l$ as input and $\widehat{\mathbf{X}}_{SP}^l$ as hidden states, and the output context embeddings $\mathbf{C}^{l-1} \in \mathbb{R}^{N_p \times D}$ at block $l-1$ are fed into the layer as cell states. Finally, the output prompt tokens $\mathbf{X}_P^l$ are used as the new prompts for this long-term prompt coding, which is formulated as

$$\mathbf{X}_P^l, \mathbf{C}^l = \text{LSTM}(\widehat{\mathbf{X}}_{SP}^l, \mathbf{C}^{l-1}, \mathbf{P}^l) \tag{7}$$

where $\mathbf{X}_P^l, \mathbf{C}^l \in \mathbb{R}^{N_p \times D}$ denote the updated prompts and context embeddings, respectively. $D$ denotes the dimension of embeddings, and $\mathbf{P}^l$ is new learnable parameters inserted in the block. $\text{LSTM}[\cdot]$ is the LSTM layer operator. After spatial prompt coding in block $l$, the LSTM layer takes the inserted prompts $\mathbf{P}^l$ from block $l$ as input sequences and class-ware spatial prompts $\widehat{\mathbf{X}}_{SP}^l$ from block $l$ as hidden states, and uses the previous output context $\mathbf{C}^{l-1}$ at block $l-1$ as cell states to generate the final prompt $\mathbf{X}_P^l$ as input prompt tokens to block $l+1$.

By integrating long-term coded prompts, our LSPT actively mitigates the forgetting of earned prompt tokens from previous self-attention blocks. Note that the LSTM layer for long-term temporal

coding is used starting from the first block and ending before the last block. With weights-specific LSTM layers for each block, we do not see obvious performance gains but introduce $(L-1)\times$ trainable parameters. To balance the performance and total tunable parameters, we apply one weights-shared LSTM layer for efficient visual prompt tuning.

The overall framework of our model is optimized in an end-to-end manner with class-aware spatial coding and long-term temporal coding together. By weaving in patch tokens as spatial coding elements across temporal coding, it additionally ensures a sustained aggregation of class-specific features, bolstering the model's discriminative capabilities in downstream visual classification. The class token learned from the transformer is used for the downstream classification.

## 4 EXPERIMENTS

In this section, we will introduce the experiments conducted by us to answer the following research questions:

**Q1.** How well does our LSPT perform on transfer learning benchmarks compared to the previous visual prompting baselines?

**Q2.** To what extent does the class-aware spatial prompt coding and long-term prompt coding contribute to the final performance?

**Q3.** Does the class-aware spatial prompt coding and long-term prompt coding help address the spatial and temporal forgetting problem?

### 4.1 EXPERIMENTAL SETUP

We first introduce the dataset, evaluation metrics we used and our implementation for the experiments.

**Datasets.** Our experiments are conducted on two widely used classification datasets, FGVC and VTAB-1K.

FGVC benchmark includes 5 fine-grained classification tasks: CUB-200-2011 (Wah et al., 2011), Oxford Flowers (Nilsback & Zisserman, 2008), Stanford Cars (Gebru et al., 2017), Stanford Dogs (Khosla et al., 2011), and NABirds (Van Horn et al., 2015). Following the prior work (Jia et al., 2022; Yoo et al., 2023), we use the same split for training and validation.

VTAB-1K (Zhai et al., 2019) dataset consists of 19 diverse visual classification tasks, and is composed of three subgroups: Natural with natural images obtained from standard cameras, Specialized with images captured using specialized equipments (medical and satellite imagery), and Structured which requires geometric understanding such as object counting. Each task contains 1000 training examples, and we use the same split in (Jia et al., 2022; Yoo et al., 2023) to run the final training and evaluation.

**Evaluation Metrics.** For FGVC and VTAB-1K benchmarks, we report the individual and the average accuracy on the datasets. For individual results on each benchmark, we compute the average accuracy score on the test set within three runs using different seeds. For VTAB-1K, we report the average accuracy score on three subgroups and the overall dataset.

**Implementation.** We apply ViT-B/16 as the backbone architecture in all experiments. For self-supervised vision transformers, we use MAE (He et al., 2021) and MoCo v3 (Chen et al., 2021) pre-trained on ImageNet-1K (Deng et al., 2009). We follow the same pre-trained model parameters as the prior work (Jia et al., 2022; Yoo et al., 2023).

### 4.2 COMPARISON TO PRIOR WORK

To answer **Q1** and demonstrate the effectiveness of the proposed LSPT, we comprehensively compare it to previous finetuning and visual prompt tuning baselines: 1) Linear: a vanilla baseline that used a linear layer as the classification head; 2) Adapter (Pfeiffer et al., 2020b): a strong baseline that inserted learnable MLP modules with residual connection across transformer blocks; 3) VPT (Jia et al., 2022): the first baseline that prepended learnable prompt tokens to the input sequences as

Table 1: Quantitative results of visual prompt tuning of SSL pre-trained vision transformers on FGVC datasets. Total Params denotes the total number of parameters for the backbone encoder ViT-B, prompt tokens, and the task heads.

| Method | Total Params | CUB | Flowers | Cars | Dogs | NABirds | Average |
|---|---|---|---|---|---|---|---|
| *MAE Pre-train ViT-B/16:* | | | | | | | |
| VPT-Shallow (Jia et al., 2022) | 1.02x | 42.15 | 69.15 | 43.38 | 77.07 | 57.43 | 57.84 |
| VPT-Deep (Jia et al., 2022) | 1.02x | 68.33 | 80.05 | 67.67 | 78.83 | 65.22 | 72.02 |
| GaPT (Yoo et al., 2023) | 1.02x | 70.56 | 78.55 | 71.70 | 78.90 | 67.26 | 73.39 |
| LSPT (ours) | 1.08x | **73.86** | **82.32** | **74.75** | **82.05** | **71.73** | **76.94** |
| *MoCo v3 Pre-train ViT-B/16:* | | | | | | | |
| VPT-Shallow (Jia et al., 2022) | 1.02x | 79.05 | 90.47 | 71.91 | 81.97 | 72.92 | 79.26 |
| VPT-Deep (Jia et al., 2022) | 1.02x | 82.67 | 94.41 | 79.18 | 83.33 | 75.99 | 83.12 |
| GaPT (Yoo et al., 2023) | 1.02x | 82.86 | 93.71 | 79.02 | 83.37 | 76.02 | 83.00 |
| LSPT (ours) | 1.08x | **84.29** | **95.06** | **80.12** | **84.25** | **77.16** | **84.18** |

Table 2: Quantitative results of visual prompt tuning of SSL pre-trained vision transformers on VTAB-1K benchmarks. Total Params denotes the total number of parameters for the backbone encoder ViT-B, prompt tokens, and the task heads.

| Method | Total Params | Natural (7) | Specialized (4) | Structured (8) | Average |
|---|---|---|---|---|---|
| *MAE Pre-train ViT-B/16:* | | | | | |
| Linear | 1.01x | 18.87 | 53.72 | 23.70 | 28.24 |
| Adapter | 1.17x | **54.90** | 75.19 | 38.98 | 52.47 |
| VPT-Shallow (Jia et al., 2022) | 1.01x | 39.96 | 69.65 | 27.50 | 40.96 |
| VPT-Deep (Jia et al., 2022) | 1.01x | 36.02 | 60.61 | 26.57 | 37.22 |
| GaPT (Yoo et al., 2023) | 1.01x | 47.61 | 76.86 | 36.80 | 49.22 |
| LSPT (ours) | 1.05x | 52.36 | **80.75** | **41.72** | **53.86** |
| *MoCo v3 Pre-train ViT-B/16:* | | | | | |
| Linear | 1.01x | 67.46 | 81.08 | 30.33 | 54.69 |
| Adapter | 1.22x | 74.19 | 82.66 | 47.69 | 64.82 |
| VPT-Shallow (Jia et al., 2022) | 1.01x | 67.34 | 82.26 | 37.55 | 57.94 |
| VPT-Deep (Jia et al., 2022) | 1.01x | 70.27 | 83.04 | 42.38 | 61.22 |
| GaPT (Yoo et al., 2023) | 1.01x | 74.84 | 83.38 | 49.10 | 65.80 |
| LSPT (ours) | 1.05x | **77.19** | **85.69** | **52.82** | **68.72** |

task-specific instructions for prompt tuning. Both variants (VPT-Shallow and VPT-Deep) are listed for comparison; 4) GaPT (Yoo et al., 2023): a recent and strong baseline with gated prompts and adaptive attention.

For the FGVC datasets, we report the quantitative comparison results in Table 1. As can be seen, we achieve the best results regarding all metrics for five fine-grained classification tasks compared to previous visual prompt tuning approaches using MAE and MoCo v3 pre-trained weights. In particular, the proposed LSPT superiorly outperforms GaPT (Yoo et al., 2023), the current state-of-the-art visual prompt tuning baseline, by 3.30@CUB, 3.77@Flowers, 3.05@Cars, 3.15@Dogs, and 4.47@NABirds, when evaluated on MAE pre-trained weights. Furthermore, we achieve significant performance gains compared to VPT (Jia et al., 2022), the first visual prompt tuning baseline, which indicates the importance of explicitly mitigating the forgetting of prompt tokens learned from history blocks for effective prompt tuning. These significant improvements demonstrate the superiority of our approach in visual prompt tuning.

In addition, significant gains in VTAB-1K benchmarks can be observed in Table 2. Compared to GaPT (Yoo et al., 2023), the current state-of-the-art visual prompt tuning baseline, we achieve the results gains of 2.35@Natural, 2.31@Specialized, and 3.72@Structured in terms of MoCo v3 pre-trained weights. Moreover, when evaluated on the challenging Structured datasets using MAE pre-trained weights, the proposed method still outperforms GaPT (Yoo et al., 2023) by 4.92. We also achieve highly better results against Linear and VPT (Jia et al., 2022). These results demonstrate the effectiveness of our approach in learning long-term and class-aware prompts for visual prompt tuning on downstream classification.

## 4.3 ABLATION STUDIES

In this section, we try to answer **Q2** and performed ablation studies to demonstrate the benefit of introducing Class-aware Spatial Prompt Coding (CSPC) and Long-term Prompt Coding (LPC)

Table 3: Ablation studies on Class-aware Spatial Prompt Coding (CSPC) and Long-term Prompt Coding (LPC).

| CSPC | LPC | CUB | Flowers | Cars | Dogs | NABirds | Natural (7) | Specialized (4) | Structured (8) |
|------|-----|------|---------|-------|-------|---------|-------------|-----------------|----------------|
| ✗ | ✗ | 68.33 | 80.05 | 67.67 | 78.83 | 65.22 | 67.34 | 82.26 | 37.55 |
| ✓ | ✗ | 76.28 | 85.23 | 72.16 | 80.51 | 70.63 | 72.36 | 83.56 | 45.15 |
| ✗ | ✓ | 78.12 | 89.17 | 75.38 | 82.75 | 73.58 | 75.18 | 84.28 | 48.39 |
| ✓ | ✓ | **84.29** | **95.06** | **80.12** | **84.25** | **77.16** | **77.19** | **85.69** | **52.82** |

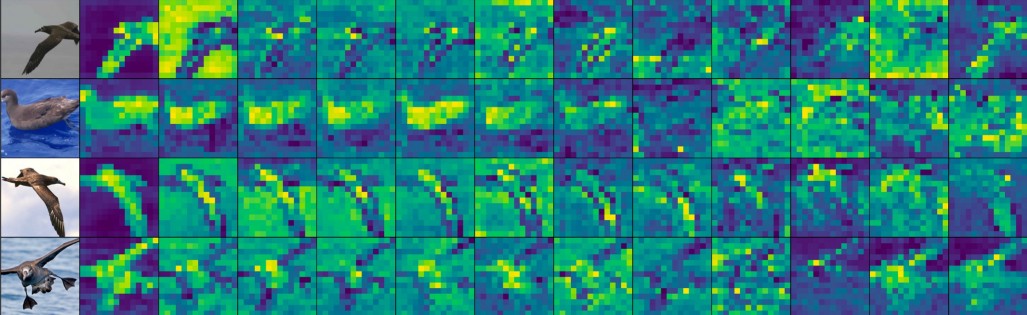

Figure 3: Qualitative visualization of long-term prompt forgetting in state-of-the-art visual prompt tuning method (Yoo et al., 2023). From left to right: layer 1 to layer 12.

modules. We ablate the necessity of each module and report the quantitative results on all downstream datasets using MoCo v3 pre-trained weights in Table 3. As can be observed, adding CSPC to the vanilla baseline highly increases the results by 9.79@CUB, 9.12@Flowers, 7.71@Cars, 3.92@Dogs, 8.36@NABirds, 7.84@Natural, 2.02@Specialized, and 10.84@Structured, which validates the benefit of long-term prompt coding in learning long-range blocks as prompt sources for visual prompt tuning. Similarly, introducing only CSPC in the baseline increases the prompt tuning performance regarding all metrics. More importantly, incorporating both LPC and CSPC into the baseline significantly raises the performance by 15.96@CUB, 15.01@Flowers, 12.45@Cars, 5.42@Dogs, 11.94@NABirds, 9.85@Natural, 3.43@Specialized, and 15.27@Structured. These improving results validate the parallel importance of mitigating the forgetting of prompt tokens learned from history blocks temporally and image patch tokens spatially for visual prompt tuning.

### 4.4 VISUALIZATION OF ATTENTION MAPS

We introduce the temporal and spatial forgetting problem in the state-of-the-art VPT method and answer **Q3** by visualizing the similarity between prompt tokens and the patch tokens as well as the attention maps in the blocks. With our Long-term Prompt Coding and Class-aware Spatial Prompt Coding, we expect to see a clear awareness of the target object from the visualization even in the very posterior blocks.

**Long-term and Spatial Prompt Forgetting in State-of-the-art VPT.** In order to validate long-term and spatial prompt forgetting in GaPT (Yoo et al., 2023), the state-of-the-art visual prompt tuning approach, we compute the cosine similarity between learnable prompt tokens and patch embeddings, and average the results along the number of prompt tokens. The qualitative visualization maps across 12 transformer layers are showcased in Figure 3, which corresponds to the temporal forgetting across blocks. As can be seen in the first column, prompt tokens in the first layer attend to learn features corresponding to objects in the image. However, prompt tokens in the last few layers fail to capture objects in the image as they do not explicitly mitigate the forgetting of prompt tokens learned from history blocks. Meanwhile, we visualize the spatial attention maps on the averaged head from 12 self-attention layers in Figure 4, which corresponds to the spatial forgetting on each transformer block. We can also observe that the spatial attention maps become worse when it comes to the last few layers since there is no direct connection between the learnable prompt tokens and local patch embeddings.

**Learned Category-aware Attention Maps.** Learning category-aware attention maps during transfer learning is essential for classifying fine-grained images. To better evaluate the quality of learned

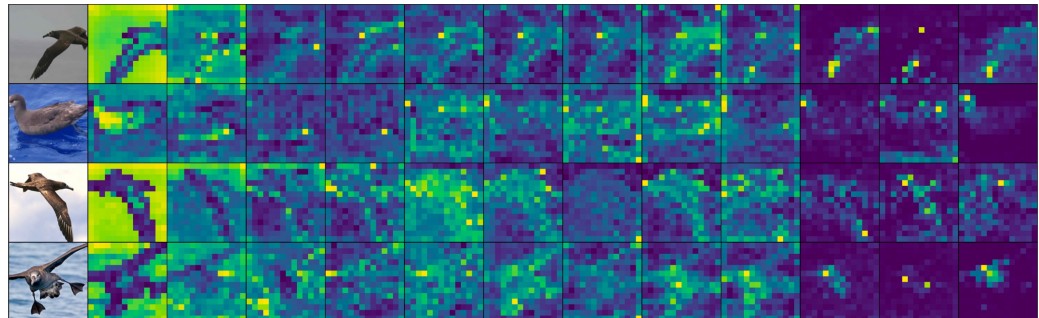

Figure 4: Qualitative visualization of spatial attention forgetting in state-of-the-art visual prompt tuning method (Yoo et al., 2023). From left to right: layer 1 to layer 12.

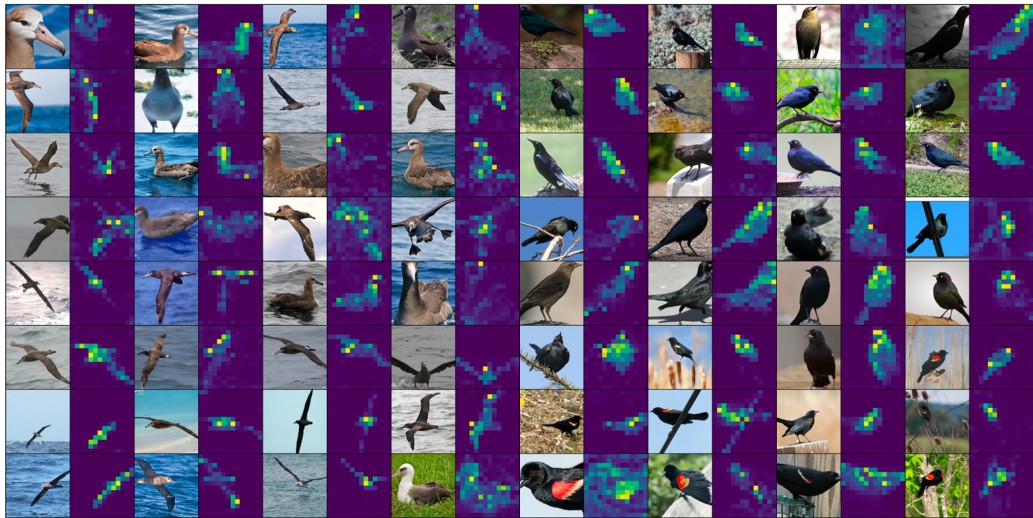

Figure 5: Qualitative visualization of learned category-aware attention maps learned by the proposed LSPT.

category-aware attention maps, we visualize the learned attention maps from the last self-attention layer by averaging all heads in Figure 5. As can be seen in the maps neighboring to the original image, attention maps extracted from our LSPT are discriminative to capture the shape of corresponding objects in the images. In contrast to our discriminative maps, the spatial attention maps from GaPT (Yoo et al., 2023) in the last column of Figure 4 are blurred and coarse, where the mixture of objects and background patches still exist. These meaningful visualization results further showcase the superiority of our LSPT in alleviating the forgetting of history prompt tokens obtained from self-attention transformer blocks for visual prompt tuning.

## 5 CONCLUSION

In this work, we navigated the multifaceted landscape of visual representation learning, pinpointing the existing gaps and challenges that have persisted, particularly in the realm of Visual Prompt Tuning (VPT). Our observations laid bare the limitations of extant methods, which, despite their efficacy, struggled to exploit the rich information embedded in long-range blocks and patch tokens of self-supervised Vision Transformers. Rising to address this need, we introduced Long-term Spatial Prompt Tuning (LSPT) with the Long-Term Prompt Coding and the Class-aware Spatial Prompt Coding which firmly establishes LSPT's capabilities in both retaining pivotal parameters from earlier blocks and continually aggregating class-centric features. The comprehensive evaluations on diverse benchmarks underscore LSPT's unmatched prowess, as it consistently outperformed prevailing baselines, setting new standards for visual prompt tuning performance.

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

## APPENDIX

In this appendix, we further provide the following materials:

- additional experiments on ADE20K semantic segmentation and supervised ImageNet-21k in Section A,
- the pseudo-algorithm framework of our LSPT in Section B,
- additional analyses on the number of LSTM & GRU layers, Long-term Prompt Coding, and Class-aware Spatial Prompt Coding in Section C,
- discussions on limitations and broader impact in Section D.

## A  ADDITIONAL EXPERIMENTS

In order to further demonstrate the effectiveness of the proposed LSPT in visual prompt tuning, we conduct experiments on semantic segmentation on ADE20 (Zhou et al., 2017; 2018) benchmark and image classification using supervised ImageNet-21K ViT-B/16 models on VTAB-1K datasets.

### A.1  ADE20K SEMANTIC SEGMENTATION

Pre-trained vision transformers are designed to be applied on a variety of downstream applications. Besides image classification, we would like to see whether our proposed LSPT can also benefit other tasks such as semantic segmentation.

To this end, we follow previous work (Jia et al., 2022; Yoo et al., 2023) and train SETR-PUP (Zheng et al., 2021) model as the segmentation transformer framework on ADE20K dataset (Zhou et al., 2017; 2018). Table A.1 reports the comparison results with previous visual prompt tuning approaches (Jia et al., 2022; Yoo et al., 2023) using MAE and MoCo v3 pre-trained ViT-B/16 weights. As can be seen, our LSPT achieves the best performance in terms of all metrics for two different pre-trained models. These significant improvements demonstrate the superiority of our framework in visual prompting on semantic segmentation.

Table A.1: Quantitative results of visual prompt tuning of SSL pre-trained vision transformers on ADE-20K for semantic segmentation. SS and MS denote single-scale and multi-scale, respectively.

| Method | MAE | | MoCo v3 | |
|---|---|---|---|---|
| | mIoU (SS) | mIoU (MS) | mIoU (SS) | mIoU (MS) |
| VPT-Shallow (Jia et al., 2022) | 34.20 | 35.23 | 34.55 | 36.18 |
| VPT-Deep (Jia et al., 2022) | 37.76 | 38.80 | 35.50 | 37.15 |
| GaPT (Yoo et al., 2023) | 38.44 | 39.81 | 36.81 | 38.55 |
| LSPT (ours) | **39.72** | **41.51** | **37.92** | **39.73** |

**Supervised ImageNet-21k.** To validate the generalizability of the proposed LSPT on visual prompt tuning using supervised weights, we comprehensively compare it with current visual prompt tuning approaches (Das et al., 2023; Wang et al., 2023; Jie et al., 2023) in supervised settings of ImageNet-21K ViT-B/16 models on VTAB-1K benchmarks (Zhai et al., 2019). The comparison quantitative results are shown in Table A.2. Compared to previous methods, we achieve the best results regarding all various benchmarks, including natural, specialized, and structured. In particular, the proposed LSPT outperforms Bi-AdaptFormer (Jie et al., 2023), the state-of-the-art method by minimizing prompts quantization errors, by 3.15@Natural, 2.17@Specialized, 3.82@Structured. We also achieve highly better results against EXPRES (Das et al., 2023) that tried to learn residual tokens for the output of various computations. These results validate the effectiveness of our approach in visual prompt tuning on supervised weights.

## B  PSEUDO ALGORITHM FOR LSPT

To enhance the replicability of our method, we report the pseudo algorithm of our LSPT in Algorithm A.1. Specifically, we are given input patch tokens for $l$th self-attention block $\mathbf{X}^{l-1} \in \mathbb{R}^{N \times D}$,

Table A.2: Quantitative results of visual prompt tuning of supervised ImageNet-21K ViT-B/16 weights on VTAB-1K benchmarks. Numbers in (·) denote the number of downstream datasets.

| Method | Natural (7) | Specialized (4) | Structured (8) | Average |
|---|---|---|---|---|
| VPT-Shallow (Jia et al., 2022) | 76.81 | 79.66 | 46.98 | 64.85 |
| VPT-Deep (Jia et al., 2022) | 78.48 | 82.43 | 54.98 | 69.42 |
| EXPRES (Das et al., 2023) | 79.70 | 84.00 | 55.00 | 70.21 |
| SNF (Wang et al., 2023) | 83.79 | 86.13 | 59.61 | 74.10 |
| Bi-AdaptFormer (Jie et al., 2023) | 82.11 | 86.40 | 62.43 | 74.73 |
| LSPT (ours) | **85.26** | **88.57** | **66.25** | **77.95** |

---

**Algorithm A.1** Long-term Spatial Prompt Tuning

---

**Require:** input patch tokens for $l$th self-attention block $\mathbf{X}^{l-1} \in \mathbb{R}^{N \times D}$, classification tokens $\mathbf{x}_C^{l-1} \in \mathbb{R}^{1 \times D}$, inserted prompt tokens $\mathbf{P}^l \in \mathbb{R}^{N_p \times D}$, $l$th ViT blocks AttnBlock$^l(\cdot)$, a single LSTM layer LSTM$(\cdot)$, number of patch tokens $N$, number of prompt tokens $N_p$, number of blocks $L$.

**for** $l = 1, 2, ..., L$ **do**

$\quad \mathbf{x}_C^l, \hat{\mathbf{X}}_P^l, \mathbf{X}^l \leftarrow \text{AttnBlock}^l(\mathbf{x}_C^{l-1}, \mathbf{X}_P^{l-1}, \mathbf{X}^{l-1})$

$\quad \hat{\mathbf{X}}_{SP,k}^l \leftarrow \hat{\mathbf{X}}_{P,k}^l + \dfrac{\sum_{i=1}^N \mathbf{X}_i^l}{N} \qquad\qquad\qquad \triangleright$ Class-aware Spatial Prompt Coding

$\quad \mathbf{X}_P^l, \mathbf{C}^l \leftarrow \text{LSTM}(\hat{\mathbf{X}}_{SP}^l, \mathbf{C}^{l-1}, \mathbf{P}^l) \qquad\qquad \triangleright$ Long-term Prompt Coding

**return** $\mathbf{X}_P^l$

---

classification tokens $\mathbf{x}_C^{l-1} \in \mathbb{R}^{1 \times D}$, inserted prompt tokens $\mathbf{P}^l \in \mathbb{R}^{N_p \times D}$, $l$th ViT blocks AttnBlock$^l(\cdot)$, a single LSTM layer LSTM$(\cdot)$, where $N, N_p, L$ denote the number of patch tokens, number of prompt tokens, and number of blocks, respectively. For each transformer block $l$, we first apply ViT self-attention blocks AttnBlock$^l(\cdot)$ to aggregate input classification tokens $\mathbf{x}_C^{l-1}$, prompt tokens $\mathbf{X}_P^{l-1}$, and patch tokens $\mathbf{X}^{l-1}$ for generating the output classification tokens $\mathbf{x}_C^l$, prompt tokens $\hat{\mathbf{X}}_P^l$, and and patch tokens $\mathbf{X}^l$.

After aggregation, we simply apply two proposed modules including the Class-aware Spatial Prompt Coding (CSPC), and Long-term Prompt Coding (LPC) on the output tokens $\hat{\mathbf{X}}_P^l, \mathbf{X}^l$, newly inserted prompt tokens $\mathbf{P}^l$, and context tokens $\mathbf{C}^{l-1}$ from last layer. The Class-aware Spatial Prompt Coding (CSPC) module adds the average embeddings of patch tokens $\mathbf{X}^l \in \mathbb{R}^{N \times D}$ from the block to the output prompts $\hat{\mathbf{X}}_P^l \in \mathbb{R}^{N_p \times D}$ to generate class-aware spatial prompts $\hat{\mathbf{X}}_{SP}^l$. With the inserted prompt tokens $\mathbf{P}^l \in \mathbb{R}^{N_p \times D}$ and $\hat{\mathbf{X}}_{SP}^l \in \mathbb{R}^{N_p \times D}$, the Long-term Prompt Coding (LPC) module with parallel importance takes the inserted prompts $\mathbf{P}^l$ as input and $\hat{\mathbf{X}}_{SP}^l$ as hidden states, and the output context embeddings $\mathbf{C}^{l-1} \in \mathbb{R}^{N_p \times D}$ at block $l-1$ are fed into the layer as cell states. Finally, the output updated prompts $\mathbf{X}_P^l$ are used as the new prompt tokens for block $l+1$ to achieve long-term prompt coding.

## C  ADDITIONAL ANALYSIS

In this section, we performed ablation studies to demonstrate the advantage of using one single LSTM layer in the Long-term Prompt Coding (LPC) module against Gated Recurrent Unit (GRU) and transformers, the benefit of adding average patch tokens in Class-aware Spatial Prompt Coding (CSPC) module, and computational costs for training and inference. Our ablation experiments are based on MAE pre-trained ViT-B/16 models.

### C.1  ABLATION ON NUMBER OF LSTM & GRU LAYERS

To validate the effectiveness of using a single LSTM layer as the Long-term Prompt Coding (LPC) module, we varied the number of LSTM layers from $\{1, 2\}$ and ablated the layer using a single Gated Recurrent Unit (GRU) layer. The comparison results are reported in Table C.1. We can observe

Table C.1: Ablation studies on Long-term Prompt Coding (LPC) regarding the number of LSTM layers and GRU.

| LPC | Params | CUB | Flowers | Cars | Dogs | NABirds | AVG |
|---|---|---|---|---|---|---|---|
| 1 # LSTM | 1.08x | 73.86 | 82.32 | 74.75 | 82.05 | 71.73 | 76.94 |
| 2 # LSTM | 1.14x | **74.57** | **82.95** | **75.52** | **82.97** | **72.45** | **77.69** |
| 1 # GRU | **1.06x** | 72.95 | 81.53 | 73.91 | 81.26 | 70.97 | 76.12 |

that adding one more LSTM layer to our current LSPT achieves better results, which indicates the importance of LSTM in alleviating long-term forgetting problems for visual prompt tuning. However, two LSTM layers bring more tunable parameters on computational overhead. Meanwhile, replacing one single LSTM layer with a single GRU layer will not improve the performance although it has fewer parameters. These results demonstrate the effectiveness of using one single LSTM layer in achieving a good trade-off between parameters and performance.

## C.2 ABLATION ON LSTM VS TRANSFORMER IN LONG-TERM PROMPT CODING

In order to further demonstrate the effectiveness of using LSTM for long-term modeling, we ablated the Long-term Prompt Coding module by using a transformer layer to aggregate previous and new prompt tokens. Table C.2 reports the comparison results on 5 fine-grained visual classification datasets. As can be observed, replacing LSTM with a transformer layer deteriorates the results in terms of all benchmarks. This might be because the auto-aggressive transformer layer applied among tokens increases their similarity by a weighted sum of token values, leading to losing discrepancy among latent toke embeddings during training. In contrast, the choice of LSTM for long-term modeling helps introduce a forget gate to enlarge the variance for prompt tokens.

Table C.2: Ablation studies on Long-term Prompt Coding (LPC) using LSTM vs Transformer.

| LPC | CUB | Flowers | Cars | Dogs | NABirds | AVG |
|---|---|---|---|---|---|---|
| Transformer | 72.52 | 81.26 | 73.58 | 81.23 | 70.86 | 75.89 |
| LSTM | **73.86** | **82.32** | **74.75** | **82.05** | **71.73** | **76.94** |

## C.3 ABLATION ON CLASS-AWARE SPATIAL PROMPT CODING

Adding the average of aggregated patch tokens from attention blocks is beneficial for accumulating spatial and positional information from self-supervised ViT attention weights. To explore such effects more comprehensively, we replaced the average operator with a k-means clustering on input patch tokens and report the quantitative results in Table C.3. Specifically, $k$ is set to match the number of visual prompts, and the centroids of clusters are added to prompt tokens as Class-aware Spatial Prompt Coding (CSPC). Our LSPT with k-means clustering achieves better results, which demonstrates the importance of spatial prompt coding in visual prompt tuning on specific downstream tasks. This advanced mechanism allows for a more detailed understanding of spatial relationships within the data, which is particularly beneficial for tasks that require fine-grained differentiation between classes. However, the additional computational cost will be taken on the k-means clustering for training time. Therefore, more exploration space on this spatial prompt coding will leave for future work.

Table C.3: Ablation studies on Class-aware Spatial Prompt Coding (CSPC) using k-means.

| Spatial Prompt | CUB | Flowers | Cars | Dogs | NABirds | AVG |
|---|---|---|---|---|---|---|
| Average | 73.86 | 82.32 | 74.75 | 82.05 | 71.73 | 76.94 |
| k-means | **74.32** | **82.56** | **74.87** | **82.23** | **71.86** | **77.17** |

## C.4 TRAINING & INFERENCE COSTS

In order to comprehensively assess the efficiency of the proposed LSPT, we compared it with GaPT Yoo et al. (2023), the state-of-the-art visual prompt tuning method on self-supervised ViTs, on max memory usage, training time per batch and inference time per batch in Table C.4. We can observe that our LSPT achieves comparable computation costs in terms of all metrics, especially on inference time per batch. More importantly, we achieve much better downstream performance regarding image classification in Table 1& 2 and semantic segmentation in Table A.1. These computational analyses further demonstrate the efficiency of our novel framework.

Table C.4: Comparsion results of training & inference costs with the state-of-the-art visual prompt tuning approach on self-supervised ViT.

| Method | Max Memory Usage (GB) | Training Time per Batch (s) | Inference Time per Batch (s) |
|---|---|---|---|
| GaPT (Yoo et al., 2023) | 23.78 | 0.2406 | 0.0871 |
| LSPT (ours) | 24.02 | 0.2428 | 0.0872 |

## D LIMITATIONS & BROADER IMPACT

LSPT is a preliminary work to address the temporal forgetting problem and the spatial forgetting problem within the visual prompt tuning techniques. While we compared several designs for long-term prompt coding and class-aware spatial prompt coding in Appendix C, most choices are rather effective yet straight-forward. However, we believe there is still much room to obtain a better balance between cost and effectiveness, which we leave for future exploration.

Since prompt tuning is a widely used concept shared across modalities, potential future directions include exploring language-guided VPT for cross-modal understanding where both images and languages are involved. This may lead to a more holistic view of LSPT, extend its potential applications and areas for wider research community.

