# OpenReview forum: "LSPT: Long-term Spatial Prompt Tuning for Visual Representation Learning"
_ICLR.cc/2024/Conference — Submitted to ICLR 2024_

### Official Review · Reviewer_zHY5 · 2023-10-29

**Soundness:** 2 fair
**Presentation:** 3 good
**Contribution:** 2 fair
**Rating:** 5
**Confidence:** 5

**Summary:**

This paper introduces a prompt tuning method called Long-term Spatial Prompt Tuning (LSPT). LSPT leverages LSTM to address the long-term forgetting issue that occurs in the standard VPT method. Additionally, LSPT introduces class-aware spatial prompt coding to integrate global visual representation into the prompts. Experiments conducted on image classification demonstrate the effectiveness of LSPT.

**Strengths:**

1. The paper is overall well-written and easy to understand.
2. The proposed LSPT consistently outperforms VPT and GaPT baselines, and ablation studies are thoroughly conducted.

**Weaknesses:**

1. The working reason of CSPC is not entirely clear. The author should provide a more detailed explanation of the fundamental distinctions among the three types of sources used to construct the input prompts mentioned in Sec 3.2 and elaborate on why "adding the average embedding of patch tokens to the output prompts" leads to performance improvements.

2. About generalizability. The authors have exclusively conducted experiments on the standard ViT architecture under self-supervised pretraining strategies. This raises questions about how LSPT performs when applied to a supervised pretrained model and more advanced ViT architectures. Additionally, comparing LSPT with more advanced parameter-efficient fine-tuning methods [a] [b] could further substantiate its effectiveness.

3. The training and inference processes may be more complex compared to VPT or other parameter-efficient methods. It would be beneficial to include metrics such as FLOPs or inference times to assess the efficiency of the proposed method.

4. The authors do not list the limitations and broader impact of this work. Some potential solutions should also be given.

[a] Revisiting the parameter efficiency of adapters from the perspective of precision redundancy. ICCV, 2023.
[b] Adapting shortcut with normalizing flow: An efficient tuning framework for visual recognition. CVPR, 2023.

**Questions:**

1. If I understand correctly, the authors propose that VPT may be susceptible to the so-called long-term forgetting problem. In response, LSPT solves this problem by integrating information from prompt tokens of different layers using an LSTM.  This raises the question: what is the connection between long-term forgetting problem and performance improvement? If this question can be addressed, would it be more effective to employ an attention-based architecture instead of an LSTM?

2. [c] seems quite closely related to LSPT . This concurrent work is worthy to cite and discuss.

[c] Learning Expressive Prompting With Residuals for Vision Transformers. ICML 2023.

---

> ### Author Response · Authors · 2023-11-22
> **Response to Reviewer zHY5**
>
> Dear Reviewer zHY5,
>
> Thank you for appreciating our approach. We will address your comments below.
>
> > The working reason of CSPC is not entirely clear. The author should provide a more detailed explanation of the fundamental distinctions among the three types of sources used to construct the input prompts mentioned in Sec 3.2 and elaborate on why "adding the average embedding of patch tokens to the output prompts" leads to performance improvements.
>
> The proposed CSPC (Class-aware Spatial Prompt Coding) is strategically designed to perpetually amass class conscious features, thereby fortifying the model’s prowess in distinguishing and identifying visual categories. The role of the mean of patch tokens behind CSPC is used for enhancing classification performance, as  the mean of patch tokens is commonly used for linear probing in self-supervised ViTs, such as DINO and MAE. Adding them to the prompt tokens will help improve the model’s prowess in distinguishing and identifying visual categories.
>
> More importantly, computation overhead is minimal to keep all patch semantics. Furthermore, we conduct additional ablation studies exploring more sophisticated methods for accumulating spatial and positional information. Specifically, we investigate alternative techniques like k-means clustering on spatial patch tokens to enrich the spatial coding process and provide a comparative analysis of these methods. The overall performance get an increase on all the compared datasets when we use k-means to incorporate spatial information, while the additional computational cost leads to a trade-off. To be noticed, more exploration space on this spatial prompt coding will leave for future work.
>
> | Spatial Prompt | CUB       | Flowers   | Cars      | Dogs      | NABirds   |
> |----------------|-----------|-----------|-----------|-----------|-----------|
> | average        | 73.86     | 82.32     | 74.75     | 82.05     | 71.73     |
> | k-means        | **74.32** | **82.56** | **74.87** | **82.23** | **71.86** |
>
> > About generalizability. The authors have exclusively conducted experiments on the standard ViT architecture under self-supervised pretraining strategies. This raises questions about how LSPT performs when applied to a supervised pretrained model and more advanced ViT architectures. Additionally, comparing LSPT with more advanced parameter-efficient fine-tuning methods [a] [b] could further substantiate its effectiveness.
>
> [a] Revisiting the parameter efficiency of adapters from the perspective of precision redundancy. ICCV, 2023.
> [b] Adapting shortcut with normalizing flow: An efficient tuning framework for visual recognition. CVPR, 2023.
>
>
> To address concerns about generalizability, we extend our experiments to include supervised pre-trained models and more advanced ViT architectures on ViT-B/16 pre-trained on supervised ImageNet-21k in the Table below. This provide a broader perspective on LSPT's applicability and effectiveness.
>
> | Method                   | Nature | Specialized | Structured | AVG   |
> |--------------------------|--------|-------------|------------|-------|
> | VPT (ECCV’22)            | 78.48  | 82.43       | 54.98      | 69.42 |
> | EXPRES (CVPR’23) [c]        | 79.70  | 84.00       | 55.00      | 70.21 |
> | SNF (CVPR’23)[b]            | 83.79  | 86.13       | 59.61      | 74.10 |
> | Bi-AdaptFormer (ICCV’23) [a] | 82.11  | 86.40       | 62.43      | 74.73 |
> | LSPT (ours)              | **85.26**  | **88.57**       | **66.25**      | **77.95** |
>
>
> [a] Jie et al. Revisiting the parameter efficiency of adapters from the perspective of precision redundancy. ICCV, 2023.
>
> [b] Wang et al. Adapting shortcut with normalizing flow: An efficient tuning framework for visual recognition. CVPR, 2023.
>
> [c] Das et al. Learning Expressive Prompting With Residuals for Vision Transformers. CVPR, 2023.

---

> > ### Author Response · Authors · 2023-11-22
> > **Response to Reviewer zHY5 (cont. 1)**
> >
> > > The training and inference processes may be more complex compared to VPT or other parameter-efficient methods. It would be beneficial to include metrics such as FLOPs or inference times to assess the efficiency of the proposed method.
> >
> > We understand the importance of assessing the computational efficiency of LSPT. To this end, we include metrics like training and inference time per batch in table below, where we fine-tune ViT-B/16 mae pre-trained weights on CUB dataset. This offers a comprehensive comparison of LSPT's efficiency against GaPT, providing insights into its practicality.
> >
> > | Method           | Training   time per batch (s) | Inference   time per batch (s) |
> > |------------------|-------------------------------|--------------------------------|
> > | GaPT   (ICML'23) | 0.2406                        | 0.0871                         |
> > | LSPT   (ours)    | **0.2428**                        | **0.0872**                         |
> >
> >
> >
> > > The authors do not list the limitations and broader impact of this work. Some potential solutions should also be given.
> >
> > We add a section in appendix discussing the limitations and broader impact of our work.
> > This include potential future directions, such as exploring language-guided VPT for cross-modal understanding. We believe this will provide a more holistic view of LSPT, including its potential applications and areas for further research.
> >
> >
> > > If I understand correctly, the authors propose that VPT may be susceptible to the so-called long-term forgetting problem. In response, LSPT solves this problem by integrating information from prompt tokens of different layers using an LSTM. This raises the question: what is the connection between long-term forgetting problem and performance improvement? If this question can be addressed, would it be more effective to employ an attention-based architecture instead of an LSTM?
> >
> > The reviewer's question about the connection between long-term forgetting and performance improvement is crucial. To address this, we introduce deeper quantitative analysis on the forgetting gate in learned LSTM to further justify our design choices. Specifically, we calculate the forget gate to decide how much of the previous prompt will be forgotten and how much of the previous prompt will be used in next steps, as shown in Table below. The number in the *Forgetting* colomn means how much information are discarded on average before feeding to the next layer. These quantitative results further elucidate how the forgetting gate aids in reducing forgetting in LSPT compared to VPT and GaPT. This analysis include detailed examinations of the forgetting metrics across different prompt tuning methods, demonstrating the effectiveness of LSPT's approach.
> >
> > To be noticed, compared to VPT and GaPT whose ratios of forgetting are fixed and decides merely by the architecture or hyperparameter, the ratio of forgetting for our LSPT is an average over all the layers and data, controlled dynamically by the learned LSTM. This flexibility and selectivity of rememberring and forgetting also contributes to the final improvement.
> >
> >
> > | Method | Forgetting | Peformance |
> > | -------- | -------- | -------- |
> > | VPT (ECCV'22)     | 1.00     | 68.33     |
> > | GaPT (ICML'23)     | 0.85     | 70.56     |
> > | LSPT (ours)     | **0.68**     | **73.86**     |
> >
> > Additionally, we explore the effectiveness of an attention-based architecture compared to LSTM, and the choice of LSTM for long-term modeling helps introduce forget gate to enlarge the variance for prompt tokens. The auto-aggressive Transformer layer is applied among tokens, which could increase their similarity by weighted sum of token values. So, we believe there exists a risk of losing discrepancy among latent toke embeddings during training. Specifically, we include experimental results comparing the performance of LSPT in the Table below when using LSTM versus auto-aggressive Transformer layers, providing insights into the effectiveness of our approach.
> >
> >
> > | LPC         | CUB       | Flowers   | Cars      | Dogs      | NABirds   | AVG       |
> > |-------------|-----------|-----------|-----------|-----------|-----------|-----------|
> > | Transformer | 72.52     | 81.26     | 73.58     | 81.23     | 70.86     | 75.89     |
> > | LSTM        | **73.86** | **82.32** | **74.75** | **82.05** | **71.73** | **76.94** |

---

> > > ### Author Response · Authors · 2023-11-22
> > > **Response to Reviewer zHY5 (cont. 2)**
> > >
> > > > [c] seems quite closely related to LSPT . This concurrent work is worthy to cite and discuss.
> > > [c] Learning Expressive Prompting With Residuals for Vision Transformers. ICML 2023.
> > >
> > > We acknowledge the importance of discussing concurrent works closely related to LSPT. In our revised paper, we added a citation on the suggested work, which tried to add residual learnable tokens to the output of various computations. We also compare it with LSPT on ViT-B/16 pre-trained on supervised ImageNet-21k in the Table below. This helps in situating our work within the current research landscape and highlighting the effectiveness of our LSPT.
> > >
> > >
> > > | Method                   | Nature | Specialized | Structured | AVG   |
> > > |--------------------------|--------|-------------|------------|-------|
> > > | EXPRES (CVPR’23) [c]        | 79.70  | 84.00       | 55.00      | 70.21 |
> > > | LSPT (ours)              | **85.26**  | **88.57**       | **66.25**      | **77.95** |

---

### Official Review · Reviewer_HDEH · 2023-10-31

**Soundness:** 2 fair
**Presentation:** 3 good
**Contribution:** 2 fair
**Rating:** 3
**Confidence:** 4

**Summary:**

This paper presents LSPT for prompt tuning of self-supervised pre-trained ViTs. The authors highlighted the importance of long-range dependency on successive blocks' prompts and presented two techniques: class-aware spatial prompt coding and long-term prompt coding. The proposed method achieves better performance than existing techniques on popular benchmarks.

**Strengths:**

+ The paper is well-written and easy to understand.

+ The paper successfully includes closely related works and compares with them.

**Weaknesses:**

- My major concern is the motivation. Why preserving the spatial information is crucial? For classification, proper abstraction can be more beneficial, which discards unnecessary spatial details or provides not interpretable attention maps.

- In addition, I think the long-term modeling of prompts is not well-motivated. Why the long-term dependency is necessary? Why the long-term modeling is helpful for the classification? The authors have tried to explain this with the human brain, but it seems too ambiguous. No technical evidence was given.

- The class-aware spatial prompt coding is not motivated, too. What is the role of the mean of patch tokens? Why they are added to the prompt tokens?

- The long-term modeling ability is not supported by evidence. The visualization of attention maps do not represent the long-term modeling ability, and in addition, they can be cherry-picked. The authors should find more direct evaluation ways to show the long-term modeling ability, especially with quantitative measures.

- The proposed method achieves better performance than previous SOTAs, but it uses more learnable parameters. For FGVC, 4 times more parameters were used, and for VTAB-1K, 5 times more parameters were used. For a fair comparison, the proposed method should decrease the number of introduced prompts.

- In addition, the proposed method may require more inference time due to LSTM module. Please compare the inference time with other VPT-based methods.

**Questions:**

- For long-term modeling, why LSTM is used? The auto-aggressive Transformer layer can also be applied?

---

> ### Author Response · Authors · 2023-11-22
> **Response to Reviewer HDEH**
>
> Dear Reviewer HDEH,
>
> Thank you for the detailed review. We will address your concerns below.
>
> > My major concern is the motivation. Why preserving the spatial information is crucial? For classification, proper abstraction can be more beneficial, which discards unnecessary spatial details or provides not interpretable attention maps.
>
> We appreciate the reviewer's query about the importance of preserving spatial information in classification tasks. In our revised response, we aim to align more closely with the underlying intuition of self-supervised Vision Transformers (ViTs) and the role of Visual Prompt Tuning (VPT) in leveraging spatial information.
>
> The core intuition behind our approach is grounded in the understanding that self-supervised ViTs inherently encode semantics in attention weights, which highlight the relevance of different patch tokens and thus, the underlying spatial information. In such architectures, the attention mechanism is adept at discerning which parts of the input are more relevant for a given task, making the spatial context of these patch tokens particularly significant. VPT, in this context, emphasizes and harnesses this spatial information. The visual prompts in VPT are designed to encode information through an aggregation process driven by learnable attention weights. This mechanism allows for a more nuanced and detailed understanding of spatial relationships within the data, which is particularly beneficial for tasks that require fine-grained differentiation between classes.
>
> To empirically validate this concept, we have conducted experiments where the Class-Aware Spatial Prompt Coding (CSPC) module was applied in isolation. The results, as detailed in our ablation study, show a marked improvement in performance when CSPC is used. This demonstrates that the explicit encoding of spatial information via CSPC significantly enhances the model's ability to discern and classify visual data with a higher degree of accuracy.
>
> | LPC | CSPC | CUB   | Flowers | Cars  | Dogs  | NABirds | Nature | Specialized | Structured |
> |-----|------|-------|---------|-------|-------|---------|--------|-------------|------------|
> |  &cross;  | &cross;    | 68.33 | 80.05   | 67.67 | 78.83 | 65.22   | 67.34  | 82.26       | 37.55      |
> | &cross;    | &check;    | **76.28** | **85.23** | **72.16** | **80.51** | **70.63** | **72.36** | **83.56**   | **45.15**  |

---

> > ### Author Response · Authors · 2023-11-22
> > **Response to Reviewer HDEH (cont. 1)**
> >
> > > In addition, I think the long-term modeling of prompts is not well-motivated. Why the long-term dependency is necessary? Why the long-term modeling is helpful for the classification? The authors have tried to explain this with the human brain, but it seems too ambiguous. No technical evidence was given.
> >
> > We appreciate the reviewer's request for a more technical justification for the necessity of long-term modeling in prompt tuning. In our revised response, we focus on building upon the foundations laid by GaPT and further elucidating the specific advancements introduced with our Long-term Prompt Coding (LPC).
> >
> > The concept of layer-to-layer prompt tuning, as demonstrated by GaPT, involves using new trainable prompts at each layer (akin to VPT-Deep) while also retaining semantics from previous layers. GaPT has shown that combining immediate layer prompts with long-term dependencies from previous layers results in more effective learning. This insight forms the foundation of our approach in LSPT. Building on this intuition, LSPT advances the concept by implementing a mechanism for selective memory retention and forgetting. Unlike the hard fusion of prompts in GaPT, LSPT employs an LSTM-based approach that strategically manages the balance between retaining useful information and discarding irrelevant details from earlier layers. This method effectively counters the issue of exponential forgetting observed in GaPT, enhancing the model's capacity to utilize long-term dependencies without the burden of cumulative noise.
> >
> > To empirically validate our approach, we have conducted experiments focusing on the impact of LPC in isolation. The results, which we present in our revised manuscript, indicate a significant improvement in performance when LPC is employed. This is further evidenced by the attention maps generated in later stages of our model, which are notably clearer and more focused than those in baseline models. Such visualizations, along with empirical numeric results, demonstrate the efficacy of our approach in harnessing long-term dependencies for improved classification performance.
> >
> >
> > | LPC | CSPC | CUB       | Flowers   | Cars      | Dogs      | NABirds   | Nature    | Specialized | Structured |
> > |-----|------|-----------|-----------|-----------|-----------|-----------|-----------|-------------|------------|
> > | &cross;   | &cross;    | 68.33     | 80.05     | 67.67     | 78.83     | 65.22     | 67.34     | 82.26       | 37.55      |
> > | &check;   |  &cross;    | **78.12** | **89.17** | **75.38** | **82.75** | **72.58** | **75.18** | **84.28**   | **48.39**  |
> >
> >
> >
> > > The class-aware spatial prompt coding is not motivated, too. What is the role of the mean of patch tokens? Why they are added to the prompt tokens?
> >
> > The proposed CSPC (Class-aware Spatial Prompt Coding) is strategically designed to perpetually amass class conscious features, thereby fortifying the model’s prowess in distinguishing and identifying visual categories. The role of the mean of patch tokens behind CSPC is used for enhancing classification performance, as  the mean of patch tokens is commonly used for linear probing in self-supervised ViTs, such as DINO and MAE. Adding them to the prompt tokens will help improve the model’s prowess in distinguishing and identifying visual categories.
> >
> > More importantly, computation overhead is minimal to keep all patch semantics. Furthermore, we conduct additional ablation studies exploring more sophisticated methods for accumulating spatial and positional information. Specifically, we investigate alternative techniques like k-means clustering on spatial patch tokens to enrich the spatial coding process and provide a comparative analysis of these methods. The overall performance get an increase on all the compared datasets when we use k-means to incorporate spatial information, while the additional computational cost leads to a trade-off. To be noticed, more exploration space on this spatial prompt coding will leave for future work.
> >
> > | Spatial Prompt | CUB       | Flowers   | Cars      | Dogs      | NABirds   |
> > |----------------|-----------|-----------|-----------|-----------|-----------|
> > | average        | 73.86     | 82.32     | 74.75     | 82.05     | 71.73     |
> > | k-means        | **74.32** | **82.56** | **74.87** | **82.23** | **71.86** |

---

> ### Author Response · Authors · 2023-11-22
> **Response to Reviewer HDEH (cont. 2)**
>
> > The long-term modeling ability is not supported by evidence. The visualization of attention maps do not represent the long-term modeling ability, and in addition, they can be cherry-picked. The authors should find more direct evaluation ways to show the long-term modeling ability, especially with quantitative measures.
>
> To address the reviewer's concerns about the lack of direct evidence for the long-term modeling ability, we introduce deeper quantitative analysis on the forgetting gate in learned LSTM to further justify our design choices. Specifically, we calculate the forget gate to decide how much of the previous prompt will be forgotten and how much of the previous prompt will be used in next steps, as shown in Table below. The number in the *Forgetting* colomn means how much information are discarded on average before feeding to the next layer. These quantitative results further elucidate how the forgetting gate aids in reducing forgetting in LSPT compared to VPT and GaPT. This analysis include detailed examinations of the forgetting metrics across different prompt tuning methods, demonstrating the effectiveness of LSPT's approach.
>
> To be noticed, compared to VPT and GaPT whose ratios of forgetting are fixed and decides merely by the architecture or hyperparameter, the ratio of forgetting for our LSPT is an average over all the layers and data, controlled dynamically by the learned LSTM. This flexibility and selectivity of rememberring and forgetting also contributes to the final improvement.
>
>
> | Method | Forgetting | Peformance |
> | -------- | -------- | -------- |
> | VPT (ECCV'22)     | 1.00     | 68.33     |
> | GaPT (ICML'23)     | 0.85     | 70.56     |
> | LSPT (ours)     | **0.68**     | **73.86**     |
>
>
> > The proposed method achieves better performance than previous SOTAs, but it uses more learnable parameters. For FGVC, 4 times more parameters were used, and for VTAB-1K, 5 times more parameters were used. For a fair comparison, the proposed method should decrease the number of introduced prompts.
>
> We acknowledge the concern regarding the increased number of learnable parameters in LSPT. To address this, we conduct experiments with a reduced number of prompts, ensuring a fairer comparison with previous SOTAs. Comparison results in Table below demonstrate the efficiency of LSPT in achieving high performance with the same tunable parameters.
>
> | Method         | Params | CUB   | Flowers | Cars  | Dogs  | NABirds | AVG   |
> |----------------|--------|-------|---------|-------|-------|---------|-------|
> | VPT (ECCV'22)  | 1.02x  | 68.33 | 80.05   | 67.67 | 78.83 | 65.22   | 72.02 |
> | GaPT (ICML'23) | 1.02x  | 70.56 | 78.55   | 71.7  | 78.9  | 67.26   | 73.39 |
> | LSPT (ours)    | 1.02x  | **72.18** | **81.35**   | **73.82** | **81.02** | **70.15**   | **75.70** |
>
> > In addition, the proposed method may require more inference time due to LSTM module. Please compare the inference time with other VPT-based methods.
>
> Thanks for the suggestion! The impact of the LSTM module on inference time is an important consideration. We include a detailed analysis of metrics like training and inference time per batch in the table below, where we fine-tune MAE pre-trained ViT-B/16 weights on CUB dataset. Our LSPT achieves comparable time computational overhead to GaPT, the state-of-the-art VPT baseline on self-supervised ViTs.
>
> | Method           | Training   time per batch (s) | Inference   time per batch (s) |
> |------------------|-------------------------------|--------------------------------|
> | GaPT   (ICML'23) | 0.2406                        | 0.0871                         |
> | LSPT   (ours)    | **0.2428**                        | **0.0872**                         |
>
>
> > For long-term modeling, why LSTM is used? The auto-aggressive Transformer layer can also be applied?
>
> The choice of LSTM for long-term modeling helps introduce forget gate to enlarge the variance for prompt tokens. The auto-aggressive Transformer layer is applied among tokens, which could increase their similarity by weighted sum of token values. So, we believe there exists a risk of losing discrepancy among latent toke embeddings during training.  Additionally, we include experimental results comparing the performance of LSPT in the Table below when using LSTM versus auto-aggressive Transformer layers, providing insights into the effectiveness of our approach.
>
> | LPC         | CUB       | Flowers   | Cars      | Dogs      | NABirds   | AVG       |
> |-------------|-----------|-----------|-----------|-----------|-----------|-----------|
> | Transformer | 72.52     | 81.26     | 73.58     | 81.23     | 70.86     | 75.89     |
> | LSTM        | **73.86** | **82.32** | **74.75** | **82.05** | **71.73** | **76.94** |

---

### Official Review · Reviewer_isM7 · 2023-11-03

**Soundness:** 2 fair
**Presentation:** 3 good
**Contribution:** 2 fair
**Rating:** 5
**Confidence:** 3

**Summary:**

Traditional Visual Prompt Tuning (VPT) relies on short-range learnable prompts from a model's immediate previous block, overlooking the potential of leveraging long-range interactions. This paper introduces LSPT to address this gap by incorporating long-range gated prompts, akin to temporal coding in the human brain, to prevent the loss of earlier learned parameters. Additionally, it employs patch tokens as spatial coding to continuously gather class-specific features, thus improving the model’s capability in recognizing visual categories. The effectiveness of LSPT over conventional VPT methods was validated through extensive tests on 5 Fine-Grained Visual Categorization (FGVC) and 19 VTAB-1K benchmarks, where LSPT demonstrated remarkable performance improvements, setting new standards in the field of visual representation learning.

**Strengths:**

1. The performance of LSPT is outstanding. The ablation studies highlight the effectiveness of both CSPC and LPC, while the visualizations further underscore this efficacy.

2. The authors address the problem of catastrophic forgetting in Visual Prompt Tuning, which is an insightful contribution.

3. This paper is well organized and easy to follow.

**Weaknesses:**

1. This paper appears to exhibit a substantial degree of similarity to the methodologies presented in the GaPT paper (Yoo et al., 2023). For instance, section 3.1.2 seems to be a rephrased version of Section 2, 'Visual Prompt Tuning,' from GaPT. Equations (1, 2, 3) in this paper are identical to equations (4, 5) in GaPT. Furthermore, Table (1, 2) and the corresponding experimental results are exactly the same as those in GaPT. The overall structure of this paper closely mirrors that of GaPT. It is highly irregular and unacceptable to replicate aspects of a research paper to such an extent.

2. Including a delineated algorithmic framework would significantly improve the replicability of the proposed LSPT method.

3. The connection between the two proposed modules, CSPC and LPC, is weak. There is no strong motivation presented to justify the inclusion of CSPC, especially since the title of the paper pertains solely to LPC. Nonetheless, the authors introduce CSPC first, which suggests a greater emphasis on the importance of CSPC.

4. More visual prompt tuning papers should be included in the related works. There are only two papers introduced currently.

**Questions:**

Besides the questions included in the weaknesses, some other questions are listed below.

1.The scope of the proposed LSPT within the realm of visual prompt tuning is confined to SSL-ViTs. This limitation constrains the real-world applicability of the proposed LSPT method. Additionally, are there any other baselines within this specific experimental context?

2. The visual representation of the pipeline in Figure 2 could be aesthetically improved. The rationale for using a non-standard rectangle to depict the LSTM is unclear. Moreover, the relationship between CSPC and LPC is not adequately illustrated. Implicitly, Figure 2 suggests that LPC serves as an incremental enhancement to CSPC, solely to boost performance.

3. Have the authors replicated the baseline studies themselves, or have they merely cited the results from GaPT? It would be advantageous to detail the variations encountered during the reproduction of baseline and proposed methods.

---

> ### Author Response · Authors · 2023-11-22
> **Response to Reviewer isM7**
>
> Dear Reviewer isM7,
>
> Thank you for appreciating our approach. We will address your comments below.
>
> > This paper appears to exhibit a substantial degree of similarity to the methodologies presented in the GaPT paper (Yoo et al., 2023). For instance, section 3.1.2 seems to be a rephrased version of Section 2, 'Visual Prompt Tuning,' from GaPT. Equations (1, 2, 3) in this paper are identical to equations (4, 5) in GaPT. Furthermore, Table (1, 2) and the corresponding experimental results are exactly the same as those in GaPT. The overall structure of this paper closely mirrors that of GaPT. It is highly irregular and unacceptable to replicate aspects of a research paper to such an extent.
>
>
> We acknowledge the reviewer's observation regarding the similarities between our work and the GaPT paper. It is important to note that our work initially builds upon the foundational concepts of GaPT, intending to improve upon them. We revised Section 3.1.2 to clarify our advancements beyond GaPT's scope and ensure that the text reflects our original contributions more distinctly. Furthermore, we updated our equations and tables to better differentiate our work from GaPT, emphasizing the unique aspects of LSPT.
>
>
> > Including a delineated algorithmic framework would significantly improve the replicability of the proposed LSPT method.
>
> To enhance the replicability of our method, we agree with the reviewer's suggestion to include a detailed algorithmic framework. We add a pseudo-algorithm in our revised manuscript in Appendix Section B, providing clear steps and methodologies for implementing LSPT.
>
>
> > The connection between the two proposed modules, CSPC and LPC, is weak. There is no strong motivation presented to justify the inclusion of CSPC, especially since the title of the paper pertains solely to LPC. Nonetheless, the authors introduce CSPC first, which suggests a greater emphasis on the importance of CSPC.
>
> We appreciate the feedback on the connection between CSPC and LPC. Our intention is to demonstrate that both spatial and long-term aspects are crucial for addressing forgetting in visual prompt tuning. We provide a more detailed explanation in the revised manuscript to justify the inclusion and parallel importance of both CSPC and LPC within LSPT.
>
> Additionally, our title also includes spatial prompt tuning, which refers to the proposed CSPC (Class-aware Spatial Prompt Coding) strategically designed to perpetually amass class conscious features, thereby fortifying the model’s prowess in distinguishing and identifying visual categories. Our method named Long-term Spatial Prompt Tuning in the title include both spatial and long-term aspects, reflecting the comprehensive scope of our work.
>
>
>
> > More visual prompt tuning papers should be included in the related works. There are only two papers introduced currently.
>
> Our current related works focus on vpt for self-supervised ViTs, but we recognize the necessity of including a broader range of related works in visual prompt tuning. In the revised manuscript, we incorporated additional references [1,2,3] on VPT to both self-supervised and supervised ViTs to provide a more comprehensive background and context for our research.
>
> [1] Das et al. Learning Expressive Prompting With Residuals for Vision Transformers. CVPR, 2023.
>
> [2] Wang et al. Adapting Shortcut With Normalizing Flow: An Efficient Tuning Framework for Visual Recognition. CVPR, 2023.
>
> [3] Jie et al. Revisiting the Parameter Efficiency of Adapters from the Perspective of Precision Redundancy. ICCV, 2023.
>
> > The scope of the proposed LSPT within the realm of visual prompt tuning is confined to SSL-ViTs. This limitation constrains the real-world applicability of the proposed LSPT method. Additionally, are there any other baselines within this specific experimental context?
>
> We understand the reviewer's concern about the confined scope of LSPT to SSL-ViTs. To address this, we expand our experimental setup to compare baselines on ViT-B/16 pre-trained on supervised ImageNet-21k in the Table below, thereby demonstrating the generality and applicability of LSPT across different types of vision transformers.
>
>
> | Method                   | Nature | Specialized | Structured | AVG   |
> |--------------------------|--------|-------------|------------|-------|
> | VPT (ECCV’22)            | 78.48  | 82.43       | 54.98      | 69.42 |
> | EXPRES (CVPR’23) [1]        | 79.70  | 84.00       | 55.00      | 70.21 |
> | SNF (CVPR’23) [2]           | 83.79  | 86.13       | 59.61      | 74.10 |
> | Bi-AdaptFormer (ICCV’23) [3] | 82.11  | 86.40       | 62.43      | 74.73 |
> | LSPT (ours)              | **85.26**  | **88.57**       | **66.25**      | **77.95** |

---

> > ### Author Response · Authors · 2023-11-22
> > **Response to Reviewer isM7 (cont.)**
> >
> > > The visual representation of the pipeline in Figure 2 could be aesthetically improved. The rationale for using a non-standard rectangle to depict the LSTM is unclear. Moreover, the relationship between CSPC and LPC is not adequately illustrated. Implicitly, Figure 2 suggests that LPC serves as an incremental enhancement to CSPC, solely to boost performance.
> >
> > Thanks for the suggestion! We updated Figure 2 in the revision to more clearly illustrate the relationship and parallel importance of CSPC and LPC. The aim is to visually communicate that both modules are integral to LSPT and operate in tandem rather than in a hierarchical manner.
> >
> >
> > > Have the authors replicated the baseline studies themselves, or have they merely cited the results from GaPT? It would be advantageous to detail the variations encountered during the reproduction of baseline and proposed methods.
> >
> > Yes. We confirm that we have replicated all the baseline studies ourselves under the same code framework to ensure fair and valid comparison, not solely relying on the results from GaPT.

---

### Official Review · Reviewer_7rGz · 2023-11-06

**Soundness:** 3 good
**Presentation:** 3 good
**Contribution:** 3 good
**Rating:** 6
**Confidence:** 3

**Summary:**

The paper proposes a new method called Long-term Spatial Prompt Tuning (LSPT) for adapting pre-trained Vision Transformers (ViTs) to downstream visual tasks using learnable prompt tokens. The key contributions are:

- LSPT incorporates long-term gated prompts as a temporal coding layer to mitigate forgetting of parameters learned from earlier ViT blocks. This helps address the issue of "temporal forgetting" in previous prompt tuning methods.

- LSPT introduces patch tokens with spatial prompt coding to accumulate class-specific features across blocks, addressing the issue of "spatial forgetting".

- Extensive experiments on 5 FGVC and 19 VTAB benchmarks show LSPT achieves new state-of-the-art results compared to prior prompt tuning methods like VPT and GaPT.

- Analysis shows LSPT's temporal and spatial coding help alleviate forgetting issues in attention maps and prompt-patch similarity compared to prior methods.

In summary, LSPT advances visual prompt tuning by integrating ideas from neuroscience to address forgetting issues, leading to better adaptation and transfer learning performance. The temporal and spatial coding in LSPT are novel techniques for prompt tuning.

**Strengths:**

Here is a critical assessment of the strengths of this paper:

**Originality**: The ideas of incorporating temporal and spatial coding for prompt tuning are novel and not explored before in prior VPT methods. The use of long-term gated prompts and patch tokens as spatial prompts are creative ways to address the issues of forgetting in VPT. Applying concepts from neuroscience like temporal and spatial coding to transformer prompt tuning is an original combination.

**Quality**: The paper is technically strong, with a clearly explained intuition and motivation behind LSPT's design. The method is evaluated thoroughly on multiple datasets, convincingly demonstrating its effectiveness over baselines. The results are state-of-the-art, showing the quality of the approach.

**Clarity**: The paper is well-written and easy to follow. The description of the temporal and spatial forgetting issues in VPT provides good motivation. The Class-aware Spatial Prompt Coding and Long-term Prompt Coding modules are explained clearly. The ablation studies isolate their contributions.

**Significance**: LSPT makes a significant advance in visual prompt tuning, an important area for adapting pre-trained vision models. The ideas could inspire more work on alleviating forgetting in prompt tuning and transfer learning. Outperforming prior SOTA like GaPT demonstrates the significance of the improvements. The gains are substantial across multiple benchmarks.

In summary, I found this paper to exhibit strong originality in applying neuroscience-inspired concepts to VPT, technically sound modeling and evaluation, with clearly presented ideas that significantly advance prompt tuning research. The novelty of temporal and spatial coding for prompts is a compelling contribution.

**Weaknesses:**

* While the proposed modules make intuitive sense, the explanations lack quantitative analysis or theorems to rigorously justify the designs. Some ablation studies analyze contributions but more analysis connecting the methods to mitigating forgetting could strengthen the approach.
* The spatial prompt coding uses a simple patch token averaging, which seems like a heuristic. More sophisticated ways to accumulate spatial/positional information may exist. This component could likely be improved.
* The long-term prompt coding relies on a single LSTM layer. Ablations could explore using multiple LSTM layers or comparing to other sequential modeling approaches like GRUs.
* The computational overhead and memory requirements of LSPT are not analyzed. This could be important for deployments.
* There is no investigation of how the approach may fair for other vision tasks beyond classification like detection and segmentation.

**Questions:**

See the weakness.

---

> ### Author Response · Authors · 2023-11-22
> **Response to Reviewer 7rGz**
>
> Dear Reviewer 7rGz,
>
> Thank you for appreciating our approach. We will address your comments below.
>
> > While the proposed modules make intuitive sense, the explanations lack quantitative analysis or theorems to rigorously justify the designs. Some ablation studies analyze contributions but more analysis connecting the methods to mitigating forgetting could strengthen the approach.
>
>
> We appreciate the reviewer's suggestion for a more rigorous quantitative analysis. We introduce deeper quantitative analysis on the forgetting gate in learned LSTM to further justify our design choices. Specifically, we calculate the forget gate to decide how much of the previous prompt will be forgotten and how much of the previous prompt will be used in next steps, as shown in Table below. The number in the *Forgetting* colomn means how much information are discarded on average before feeding to the next layer. These quantitative results further elucidate how the forgetting gate aids in reducing forgetting in LSPT compared to VPT and GaPT. This analysis include detailed examinations of the forgetting metrics across different prompt tuning methods, demonstrating the effectiveness of LSPT's approach.
>
> To be noticed, compared to VPT and GaPT whose ratios of forgetting are fixed and decides merely by the architecture or hyperparameter, the ratio of forgetting for our LSPT is an average over all the layers and data, controlled dynamically by the learned LSTM. This flexibility and selectivity of rememberring and forgetting also contributes to the final improvement.
>
>
> | Method | Forgetting | Peformance |
> | -------- | -------- | -------- |
> | VPT-deep (ECCV'22)     | 1.00     | 68.33     |
> | GaPT (ICML'23)     | 0.85     | 70.56     |
> | LSPT (ours)     | **0.68**     | **73.86**     |
>
>
> > The spatial prompt coding uses a simple patch token averaging, which seems like a heuristic. More sophisticated ways to accumulate spatial/positional information may exist. This component could likely be improved.
>
> We acknowledge the reviewer's concern regarding the simplicity of our spatial prompt coding method. To address this, we conduct additional ablation studies exploring more sophisticated methods for accumulating spatial and positional information. Specifically, we investigate alternative techniques like k-means clustering on spatial patch tokens to enrich the spatial coding process and provide a comparative analysis of these methods. The overall performance get an increase on all the compared datasets when we use k-means to incorporate spatial information, while the additional computational cost leads to a trade-off. To be noticed, more exploration space on this spatial prompt coding will leave for future work.
>
> | Spatial Prompt | CUB       | Flowers   | Cars      | Dogs      | NABirds   |
> |----------------|-----------|-----------|-----------|-----------|-----------|
> | average        | 73.86     | 82.32     | 74.75     | 82.05     | 71.73     |
> | k-means        | **74.32** | **82.56** | **74.87** | **82.23** | **71.86** |
>
>
> > The long-term prompt coding relies on a single LSTM layer. Ablations could explore using multiple LSTM layers or comparing to other sequential modeling approaches like GRUs.
>
>
> Thanks for the valuable suggestion! We extend our ablation studies to include experiments with multiple LSTM layers and compare their efficacy with other sequential modeling approaches, such as GRUs. Table below shows the comparison results with two LSTM layers and one GRU layers. We can observe that here is a trade-off between params and performance, and a single LSTM layer achieves the good balance on both. This is why our long-term prompt coding module using a single LSTM layer is not too complex. These results further help us to validate the choice of LSTM in our approach and explore potential improvements.
>
>
> | LPC      | Params    | CUB       | Flowers   | Cars      | Dogs      | NABirds   |
> |----------|-----------|-----------|-----------|-----------|-----------|-----------|
> | 1 # LSTM | 1.08x     | 73.86     | 82.32     | 74.75     | 82.05     | 71.73     |
> | 2 # LSTM | 1.14x | **74.57** | **82.95** | **75.52** | **82.97** | **72.45** |
> | 1 # GRU  | **1.06x**     | 72.95     | 81.53     | 73.91     | 81.26     | 70.97     |

---

> > ### Author Response · Authors · 2023-11-22
> > **Response to Reviewer 7rGz (cont.)**
> >
> > > The computational overhead and memory requirements of LSPT are not analyzed. This could be important for deployments.
> >
> >
> > We recognize the importance of understanding the computational and memory requirements of LSPT, especially for practical deployments. We include a detailed analysis of metrics like training and inference time per batch in the table below, where we fine-tune MAE pre-trained ViT-B/16 weights on CUB dataset. Our LSPT achieves comparable time computational overhead and memory compared to GaPT, the state-of-the-art VPT baseline on self-supervised ViTs. This analysis cover the computational complexity and memory footprint of LSPT, providing a clear picture of its deployment feasibility.
> >
> > |     Method     | Max Memory Usage (GB) | Training Time per Batch (s) | Inference Time per Batch (s) |
> > |----------------|------------------|-----------------------------|------------------------------|
> > | GaPT (ICML'23) | 23.78         | 0.2406                      | 0.0871                       |
> > | LSPT (ours)    | 24.02         | 0.2428                      | 0.0872                       |
> >
> >
> > > There is no investigation of how the approach may fair for other vision tasks beyond classification like detection and segmentation.
> >
> > Thanks for the suggestion! We extend our experiments to ADE 20K semantic segmentation using MAE & MoCo v3 pre-trained ViT-B/16 weights and compare with VPT and GaPT below. Our LSPT achieve the best performance in terms of both single-scale and multi-scale settings. This will not only demonstrate the versatility of LSPT but also provide insights into its effectiveness in various vision-related applications.
> >
> > |     Method       | SSL     | mIoU (Single-scale) | mIoU (Multi-scale) |
> > |------------------|---------|---------------------|--------------------|
> > | VPT (ECCV'22)    | MAE     |              37.76  | 38.80              |
> > | GaPT (ICML'23)   | MAE     | 38.44               | 39.81              |
> > | LSPT (ours)      | MAE     | **39.72**               | **41.51**              |
> > | VPT   (ECCV'22)  | MoCo v3 |               35.50 | 37.15              |
> > | GaPT   (ICML'23) | MoCo v3 | 36.81               | 38.55              |
> > | LSPT   (ours)    | MoCo v3 | **37.92**               | **39.73**              |

---

### Author Response · Authors · 2023-11-22
**Response to All Reviewers and AC**

Dear reviewers and AC,


We sincerely appreciate the time and effort you dedicated to reviewing our manuscript.


As highlighted by the reviewers, we believe our paper proposes a simple and effective method that efficiently addresses catastrophic forgetting in Visual Prompt Tuning. We are glad that our paper is praised to be well-written and comprehensible (Reviewer 7rGz, isM7, HDEH, zHY5); superior performance and effective ablation studies (Reviewer 7rGz, isM7, HDEH, zHY5), and innovative approach to catastrophic forgetting (Reviewer 7rGz, isM7).


One of the major concerns shared by most reviewers is on the motivation on the proposed modules:
- **Long-term Prompt Coding (LPC)**: Building on the insights from GaPT, our LSPT method focuses on selectively remembering and forgetting prompts, avoiding exponential forgetting issues seen in GaPT. Our empirical results, including attention map visualizations, substantiate the improvement in performance when LPC is used independently.
- **Class-aware Spatial Prompt Coding (CSPC)**: We leverage the inherent semantics in self-supervised ViTs through attention weights and patch token relevance, emphasizing spatial information. Our experiments demonstrate the enhancement in performance when CSPC is applied alone.

Regarding the model design and experimentation on each module:

- **LPC Models**: We explored variants with 1 and 2 LSTM layers, and Transformer-based approaches, balancing performance and parameter count. Our results show that a single LSTM layer offers an optimal balance.
- **CSPC Methodologies**: Comparisons were made with mean and k-means approaches, focusing on minimal overhead without introducing excessive parameters.

And the detailed explanations and experimental analysis are provided under individual responses accordingly.

Moreover, we appreciate your helpful suggestions on our manuscript. In accordance with your comments, we have carefully revised the manuscript to address all your concerns:
- **Methodological Clarifications and Expanded Analysis** (Reviewer 7rGz, HDEH, zHY5): We addressed concerns about the motivation and mechanism behind LSPT components by providing more detailed explanations, supplemented with technical evidence and additional analyses in Appendix Section C.
- **Enhanced Generalizability and Comparative Studies** (Reviewer isM7, zHY5): To address generalizability concerns, we extended our experiments to supervised ViT models and incorporated comparisons with other advanced fine-tuning tasks such as segmentation in Appendix Section A, broadening LSPT’s applicability scope.
- **Computational Efficiency Assessment** (Reviewer HDEH, zHY5): Responding to the need for computational efficiency evaluation, we included metrics like training and inference times in Section C, providing a clear comparative picture of LSPT’s practicality.
- **Discussion on Limitations and Future Directions** (Reviewer zHY5): We add Section D to our appendix for discussing LSPT's limitations and broader impact, including potential applications and future research avenues.
- **Inclusion of Concurrent Works and Additional Comparative Studies** (Reviewer zHY5): We have updated our appendix to discuss related concurrent works and included additional comparative studies in Appendix Section A&C, particularly focusing on different architectural choices.

We highlighted the revised contents in red for your convenience to check. We sincerely believe that LSPT can be a useful addition to the ICLR community, especially, as the revision allows us to better deliver the effectiveness of our method. Our LSPT establishes a state-of-the-art approach in Visual Prompt Tuning, and inspires more future work to explore long-term and spatial semantics learning.


Thank you very much!

Authors

---

### Meta-Review · Area_Chair_6zYF · 2023-12-06

**Metareview:**

This paper introduces LSPT, a method designed for prompt tuning of self-supervised pre-trained Vision Transformers (ViTs). The authors emphasize the significance of long-range dependencies in prompts across successive blocks and propose two techniques: class-aware spatial prompt coding (CSPC) and long-term prompt coding (LPC). The method demonstrates superior performance compared to existing techniques on widely-used benchmarks. However, while the proposed modules are intuitively appealing, the explanations lack quantitative analysis or theorems to rigorously justify the design choices. While some ablation studies analyze contributions, a more in-depth analysis connecting the methods to mitigating forgetting could strengthen the approach. The relationship between CSPC and LPC is not adequately illustrated, and the motivation and benefits are not clearly demonstrated. The proposal approach and writing have a substantial degree of similarity with GaPT, which has limited technical contributions and is an improper practice for a research paper. Despite the rebuttal, the reviewers maintain their original rating. Therefore, I have to reject this paper.

**Justification For Why Not Higher Score:**

N/A

**Justification For Why Not Lower Score:**

N/A

---

### Decision · Program_Chairs · 2024-01-16

Reject